# Antiferromagnetic CuMnAs multi-level memory cell with microelectronic compatibility

K. Olejník[1,*], V. Schuler[1,*], X. Marti[1], V. Novák[1], Z. Kašpar[1], P. Wadley[2], R.P. Campion[2], K.W. Edmonds[2], B.L. Gallagher[2], J. Garces[3], M. Baumgartner[4], P. Gambardella[4] & T. Jungwirth[1,2]

Antiferromagnets offer a unique combination of properties including the radiation and magnetic field hardness, the absence of stray magnetic fields, and the spin-dynamics frequency scale in terahertz. Recent experiments have demonstrated that relativistic spin-orbit torques can provide the means for an efficient electric control of antiferromagnetic moments. Here we show that elementary-shape memory cells fabricated from a single-layer antiferromagnet CuMnAs deposited on a III–V or Si substrate have deterministic multi-level switching characteristics. They allow for counting and recording thousands of input pulses and responding to pulses of lengths downscaled to hundreds of picoseconds. To demonstrate the compatibility with common microelectronic circuitry, we implemented the antiferromagnetic bit cell in a standard printed circuit board managed and powered at ambient conditions by a computer via a USB interface. Our results open a path towards specialized embedded memory-logic applications and ultra-fast components based on antiferromagnets.

[1] Institute of Physics, Academy of Sciences of the Czech Republic, Cukrovarnická 10, 162 00 Praha 6, Czech Republic. [2] School of Physics and Astronomy, University of Nottingham, Nottingham NG7 2RD, UK. [3] IGS Research, Calle La Coma, Nave 8, La Pobla de Mafumet, Tarragona 43140, Spain. [4] Department of Materials, ETH Zürich, Hönggerbergring 64, Zürich CH-8093, Switzerland. * These authors contributed equally to this work. Correspondence and requests for materials should be addressed to K.O. (email: olejnik@fzu.cz).

In ferromagnetic materials, all magnetic moments sitting on individual atoms point in the same direction and can be switched by running an electrical current through a nearby electromagnet. This is the principle of recording in ferromagnetic media used from the 19th century magnetic wire recorders to today's hard-drives. Magnetic storage has remained viable throughout its entire history and today is the key technology providing the virtually unlimited data space on the internet. To keep it viable, the 19th century inductive coils were first removed from the readout and replaced by the 20th century spin-based magneto-resistive technology[1]. Twenty first century physics brought yet another revolution by eliminating the electromagnetic induction from the writing process in magnetic memory chips and replacing it with the spin–torque phenomenon[1]. In the non-relativistic version of the effect, switching of the recording ferromagnet is achieved by electrically transferring spins from a fixed reference permanent magnet. In the recently discovered relativistic version of the spin torque[2–5], the reference magnet is eliminated and the switching is triggered by the internal transfer from the linear momentum to the spin angular momentum under the applied writing current[6]. The complete absence of electromagnets or reference permanent magnets in this most advanced physical scheme for writing in ferromagnetic spintronics has served as the key for introducing the physical concept[7] for the efficient control of magnetic moments in antiferromagnets (AFs) that underpins our work.

In their simplest form, compensated AFs have north poles of half of the microscopic atomic moments pointing in one direction and the other half in the opposite direction. This makes the external magnetic field inefficient for switching magnetic moments in AFs. Instead, our devices rely on the recently dicovered special form of the relativistic spin torque[7,8]. When driving a macroscopic electrical current through certain AF crystals whose magnetic atoms occupy inversion-partner lattice sites (for example, in AF CuMnAs or Mn$_2$Au), a local relativistic field is generated which points in the opposite direction on magnetic atoms with opposite magnetic moments. The staggered relativistic field is then as efficient in switching the AF as a conventional uniform magnetic field in switching a ferromagnet. This reverses the traditionally sceptical perception of the utility of AFs in microelectronics and opens avenues for spintronics research and applications[9–12].

In the present paper we focus on the multi-level switching characteristics of the memory bit-cells patterned into an elementary cross-shape geometry from a single metallic layer of the CuMnAs AF deposited on a III–V or Si substrate. The multiple-stability, reflecting series of reproducible, electrically controlled domain reconfigurations[13], is not favourable for maximizing the retention and the bit-cell size scalability. However, in combination with the simplicity of the bit-cell geometry and unique features of AFs stemming from their zero net moment, the multi-level nature may provide additional functionalities, such as a pulse counter, with a utility in future specialized embedded memory-logic components in the 'More than Moore'[14] internet of things (IoT) applications. The endurance, retention, and the bit-size scalability are important parameters governing the development of bistable ferromagnetic bit-cells for non-volatile magnetic random access memories (MRAMs). Outside the realm of high-density main computer memories, the requirements on these parameters might be less stringent as long as the memories have other merits suitable for the specific embedded applications. In particular, the components we perceive are multi-level AF bit-cell chips with each bit-cell integrating memory and pulse-counter functionalities.

## Results

**Overview.** In the first and second parts of the paper we focus on the response of our bit-cells to electrical pulses in the microsecond to millisecond range. To highlight the realistic

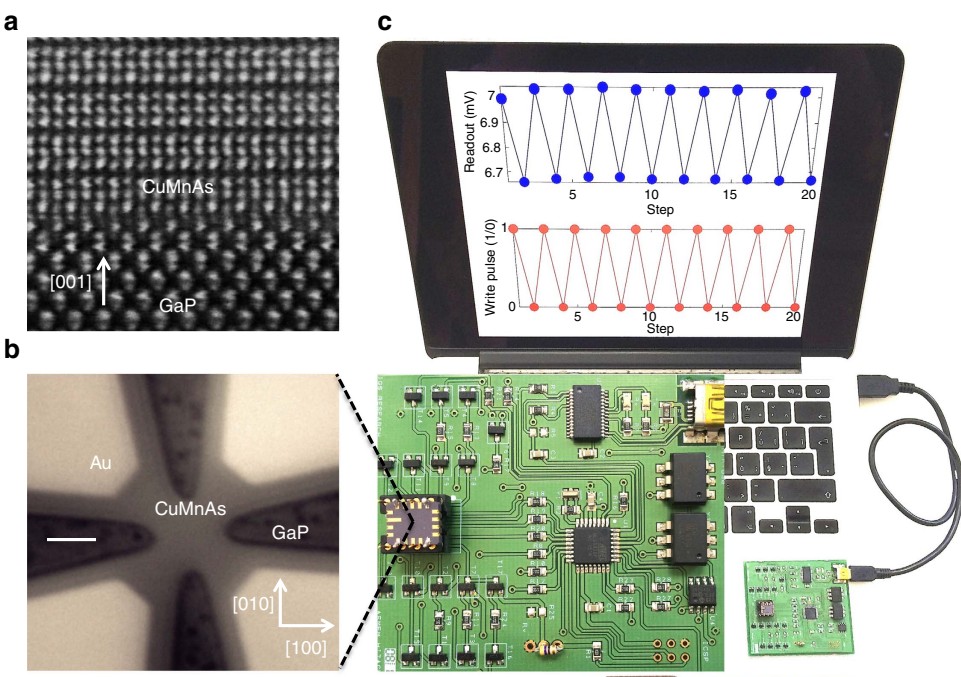

**Figure 1 | Antiferromagnetic microelectronic memory device.** (**a**) Scanning transmission electron microscopy image in the [100]–[001] plane of the CuMnAs epilayer grown on a GaP substrate. (**b**) Optical microscopy image of the device containing Au contact pads (light) and the AF CuMnAs cross-shape bit cell on the GaP substrate (dark). Scale bar length is 2 µm. (**c**) Picture of the PCB with the chip containing the AF bit cell and the input write-pulse signals (red dots) and output readout signals (blue dots) sent via a USB computer interface.

prospect of transferring the only very recent scientific discovery[8] of the electrical control of AFs from laboratory experiments to future practical IoT applications, we start in the first part by describing our implementation of the multi-level CuMnAs bit-cell in a standard printed circuit board (PCB). In the following part we present systematic data on the memory-counter characteristics as a function of the pulse length, duty cycle, and integrated pulse-time. In the third part of the paper we extend the measurements to pulse lengths scaled down to a ~100 ps range. These are the limiting pulse lengths accessible electrically and we demonstrate a reproducible memory-counter functionality with ~1,000 pulses. All combined, our elementary-shape micron-size bit cells can act as a multi-level memory-counter over the entire range of electrical pulse lengths downscaled to ~100 ps. Finally, in the discussion section, we summarize the prospects of future research and applications of the AF spin-orbit torque devices.

**Antiferromagnetic bit cell in a USB-connected device.** Figures 1 and 2 provide an overview of the basic characteristics of our AF

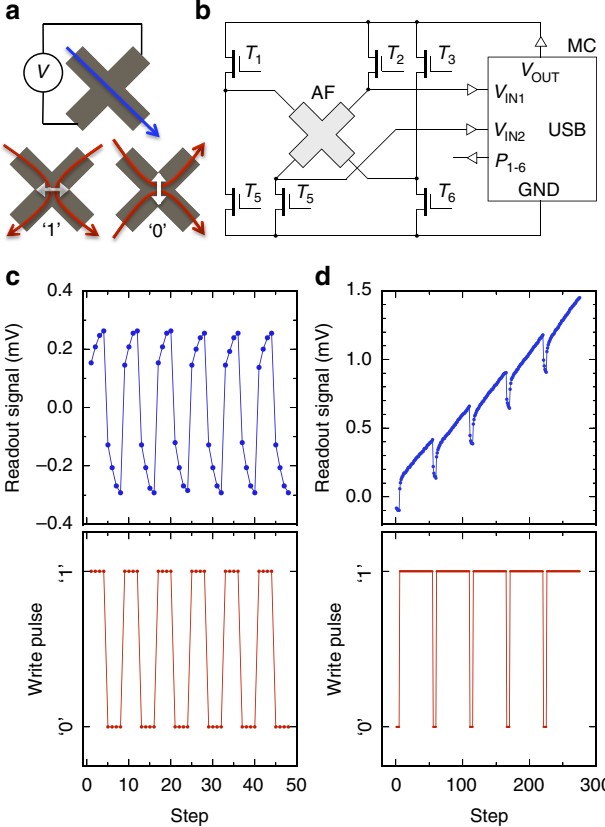

**Figure 2 | Antiferromagnetic multi-level memory bit-cell.** (**a**) The readout current (blue arrow) and transverse voltage detection geometry; write pulse current lines (red arrows) labelled '1' and '0' and the corresponding preferred AF moment orientations (white double-arrows). (**b**) Schematics of the circuitry controlling the write/read functions. Microcontroller (MC) supplies the AF bit-cell circuit through its adjustable voltage output $V_{OUT}$; different writing and reading configurations are realized by switching transistors $T_1$ to $T_6$ controlled by digital outputs $P_1$ to $P_6$ of the MC; transversal voltage is sensed differentially by analogue voltage inputs $V_{IN1}$ and $V_{IN2}$ of the MC. GND labels ground. (**c**) A symmetric pulsing with repeated four write pulses with current lines along the [100] direction labelled '0' followed by four pulses with current lines along the [010] direction labelled '1' (red dots); corresponding readout signals (blue dots). (**d**) Same as (**c**) with the four '0' write pulses followed by fifty '1' pulses. All measurements were performed at room temperature.

CuMnAs memory cells. For the purpose of the present study the cell has a cross shape, 2 μm in size (Fig. 1b), patterned by electron beam lithography and reactive ion etching from a 60 nm thick, single-crystal CuMnAs film (Fig. 1a). The material shown here was grown by molecular beam epitaxy (MBE) on a GaP substrate[15]. We recall that, besides basic research, MBE is widely used in the manufacture of microelectronic devices, in particular for mobile technologies. We also note that GaP is lattice matched to Si and that, as shown below, high quality CuMnAs films can be deposited on both GaP and Si at temperatures between 220 and 300 °C, that is, well below the CMOS circuit tolerance limit which is typically above 400 °C. Our CuMnAs films are metallic with a conductivity of $5-8 \times 10^3 \, \Omega^{-1} \, cm^{-1}$. The cell write/read signals can be sent at ambient conditions using a standard PCB connected to a personal computer via a USB interface (Fig. 1c).

Writing current pulses, depicted by red arrows in Fig. 2a, are sent through the four contacts of the bit-cell to generate current lines in the central region of the cross along either the [100] or [010] CuMnAs crystal axis. The writing current pulses give preference to domains with AF moments aligned perpendicular to the current lines[7,8], as shown schematically in Fig. 2a by the white double-arrows. Electrical readout is performed by running the probe current along one of the arms of the cross (blue arrow in Fig. 2a) and by measuring the AF transverse anisotropic magnetoresistance (planar Hall effect) across the other arm[8,16]. We note that ohmic anisotropic magnetoresistance (AMR) of comparable magnitude to our CuMnAs films[13] was also utilized in the first generation of MRAM integrated circuits using thin-film uniaxial ferromagnets and bridge formation in the read circuitry, comprising reference and storing cells, for eliminating thermal and noise effects[11,17,18].

The simplicity of the circuitry sufficient to operate the AF bit-cell is highlighted in Fig. 2b. Apart from the CuMnAs memory chip it contains only standard transistors and a microcontroller, powered by a 5 V USB 2.0 socket, for sending the write/read voltage signals. The device operates at ambient conditions and shows highly reproducible multi-level switching signals with a single readout step and no additional output data processing.

Examples of different write-pulse sequences and corresponding multi-level readout signals obtained with our proof-of-concept USB device are shown in Fig. 2c,d. In one case a symmetric pulsing was applied, repeating four pulses with current lines along the [100] direction followed by four pulses with current lines along the [010] direction. In the second case, the four pulses with current lines along the [100] direction are followed by fifty pulses with current lines along the [010] direction. The results illustrate a deterministic multi-level switching of the CuMnAs bit cell.

A complementary study performed at the Diamond Light Source directly associated the electrical switching signal in a CuMnAs cross structure with 10 μm wide arms with the AF moment reorientations within multiple domains of sub-micron dimensions[13]. In the experiment, several pairs of orthogonal, 50 ms writing pulses were applied and the corresponding domain reconfigurations were detected by means of the photoemission electron microscopy (PEEM) with contrast enabled by x-ray magnetic linear dichroism (XMLD). The observed spatially-averaged XMLD-PEEM signal correlated well with the measured AMR which also represented a magnetoresistance signal averaged over many domains. On a sub-micron scale, however, the XMLD-PEEM images showed a non-uniformity with domains responding significantly stronger or weaker to the writing pulses than the spatial average. Consistently, when several successive writing pulses were applied along the same direction, the increasing AMR signal of the multi-level bit cell again correlated well with the increased number of switched domains as observed in the XMLD-PEEM[13].

While in the large-facility XMLD-PEEM experiment only a very limited number of switchings could have been explored, Fig. 2c,d highlight on hundreds of pulses the level of electrical control that can be achieved over the multi-domain switching processes in AFs which, unlike ferromagnets[19], are insensitive to and do not generate dipolar magnetic fields. We now proceed to exploring in detail the dependencies of the readout signals on the parameters of writing pulses. The study presented below involves tens of thousands of switchings with individual pulse lengths spanning eight orders of magnitude from ~10 ms down to ~100 ps. We performed the experiments using laboratory electrical pulse generators or high frequency set-ups equipped with rf cables and the AF devices mounted on specially designed co-planar waveguide with rf access.

**Antiferromagnetic memory-counter.** We first focus on the multi-level bit cell characteristics when written by trains of pulses with the individual pulse length varied from milliseconds to microseconds. The results, summarized in Fig. 3a–d, were obtained using the following measurement protocol: Before each train of pulses (with writing current lines along the [100] direction), the cell was reset to the same initial state. The maximum length of the pulse train, including all pulses and delays between pulses, was set to 100 ms and readout was performed 5 s after the last pulse in the train. The writing current was fixed at 46 mA (corresponding to a current density of $2.7 \times 10^7$ A cm$^{-2}$) and the readout current was 500 µA.

In Fig. 3a we compare the dependencies of the readout signal on the number of pulses for different individual pulse lengths. The dependencies are highly reproducible as indicated by error bars obtained from repeated measurements for each pulse train. The AF bit cell acts as a counter of pulses whose number can be in hundreds. The separation of the readout signals for different numbers of pulses, that is, the accuracy of the pulse counting, increases with increasing individual pulse length and can reach a single-pulse resolution. The duty cycle was fixed in all measurements shown in Fig. 3a to 0.025. In Fig. 3b we show that for a given individual pulse length, the duty cycle (delay between pulses) can be varied over a broad range without affecting the readout signal of the counter.

In Fig. 3c,d we plot the readout signal dependence on the integrated pulse time, that is, on the number of pulses multiplied by the individual pulse length. Over a broad range of individual pulse lengths, the dependencies fall onto a universal curve making the AF memory cell a detector of the integrated pulse time, as shown in Fig. 3c. The universal trend breaks down for individual pulse lengths smaller than ≈50 µs. This can be explained by heating assisted spin-orbit torque switching in our devices. By a direct measurement of the heating during the pulse we observe that in the 2 µm cells the heating saturates at pulse lengths exceeding tens of µs. For these longer pulses, switching occurs at the saturated temperature, which results in the universal dependence of the readout signal on the integrated pulse time. For shorter pulses, the temperature during switching does not reach saturation and the heating decreases with decreasing pulse length which results in the lower readout signal. We note that in all measurements the temperature during switching stays at least 100 K below the CuMnAs Néel temperature ($T_N = 480$ K)[20].

An accurate detection of the integrated pulse time is feasible for tens of pulses in our measurements, as shown in Fig. 3d. For pulse

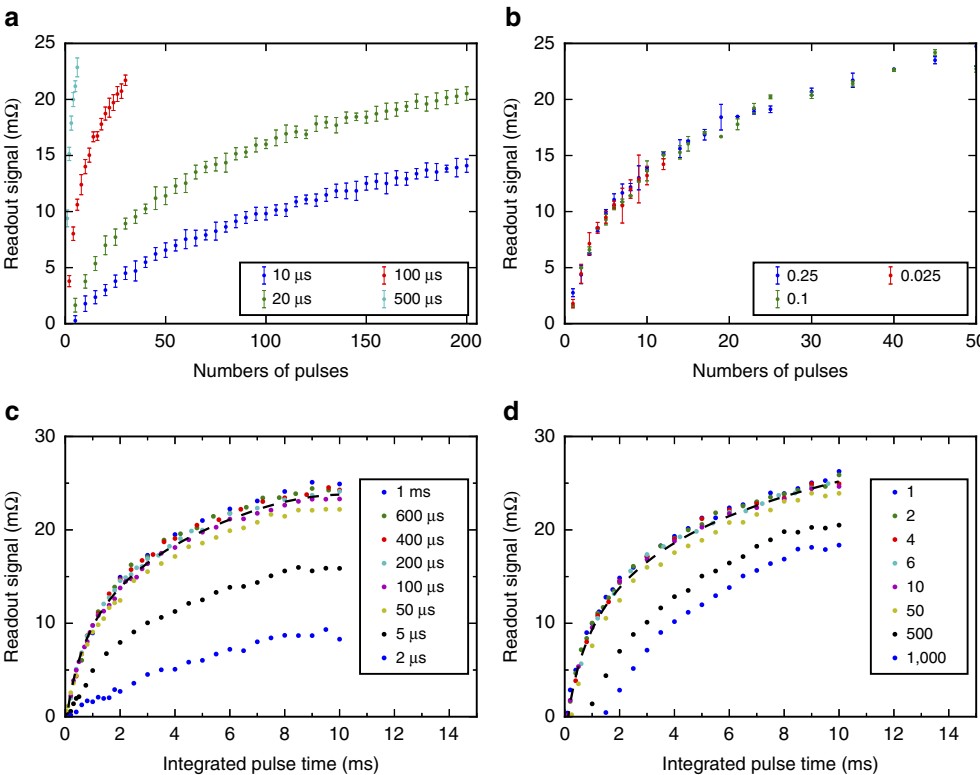

**Figure 3 | Antiferromagnetic memory-counter characteristics. (a)** Readout signal as a function of the number of pulses in the train of pulses, for different values of the individual pulse length and a common duty cycle of 0.025. All data points are obtained starting from the same reference state. The writing current density is $2.7 \times 10^7$ A cm$^{-2}$. Plotted data points are the average over ten measurements; error bars represent the standard deviation. **(b)** Same as **a** for different duty cycles (corresponding to different delays between individual pulses) and for a common individual pulse length of 200 µs. **(c)** Same as **a** measured as a function of the integrated pulse time and plotted for different individual pulse lengths. **(d)** Same as **c** plotted for different number of pulses in the pulse train. The lines connecting the data points are a guide to the eye. All measurements were performed at room temperature.

numbers exceeding one hundred, the readout signal at a given integrated pulse time drops down from the universal trend because of the non-saturated heating during the shorter pulses. The signal reduction gets stronger at lower integrated pulse times with correspondingly smaller individual pulse lengths. For the current density used in the measurements in Fig. 3, the readout signal vanishes for pulse lengths below a microsecond.

**Switching pulse lengths from tens of ms to hundreds of ps.** Our measurements show that it is possible to switch AF domains using current pulse lengths reaching the limiting, $\sim 100$ ps scale of electrical generation. This can be achieved in our AF memory cells with accessible pulse current densities while maintaining the $\sim m\Omega$ readout signal. In Fig. 4a we plot a typical dependence of the measured readout signal on the length of a single writing pulse. Before each measurement of the given pulse length, the cell was reset to the same initial state and then the single write pulse was applied with the current lines along the [100] direction. The readout signal increases with increasing pulse length. This is analogous to the dependence on the number of pulses and reflects the multi-domain nature of the switching. The initial linear increase of the readout signal with increasing pulse length defines the signal per pulse length ratio which we plot in Fig. 4b as a function of the writing current density. For comparison, we included in the plot data points for a 30 µm-size cell used in earlier measurements with pulse lengths in the range of $\sim 10^{-2}$–$10^{-3}$ s reported in ref. 8, for a 2 µm-size cell described above, and for an additional 4 µm-size cell patterned from a 50 nm thick CuMnAs epilayer on a GaAs substrate. The 30 µm cell experiments allowed us to explore only a limited range of current densities before heating damaged the sample. For the

2 and 4 µm cells, much higher current densities can be applied which allowed us to scale the writing pulse length from milliseconds[8] down to sub-nanoseconds while keeping the m$\Omega$ level of the readout signal, as illustrated in Fig. 4b. The signal per pulse length ratio shows an initial steep increase with the current density followed by a much weaker, nearly linear dependence. This is consistent with the thermally activated switching process.

In Fig. 4c, we show memory-counter measurements for individual pulse length of 250 ps, that is, at the limit of pulse lengths accessible by electrical generation. We tested counting up to 1,000 pulses and, as in the case of the $\sim$ ms and $\sim$ µs pulses, we observe a highly reproducible monotonic dependence of the readout signal on the number of pulses. Note that for individual data points the error bars were obtained from fifteen independent measurements which also means that the bit cell was exposed to $\sim 25,000$ writing pulses during this study.

Finally we illustrate in Fig. 4d that bit-cells fabricated from CuMnAs films deposited on Si at 220 °C also show the highly reproducible multi-level switching characteristics as the devices fabricated from CuMnAs on GaP or GaAs substrates (cf. with Fig. 2 and ref. 8). The plot shows an example of a symmetric pulsing experiment of repeating three writing pulses with current lines along the [100] direction followed by three pulses with current lines along the [010] direction. The corresponding histogram taken from 300 pulses highlights the deterministic switching of these multi-level CuMnAs/Si bit-cells.

## Discussion

The deterministic multi-level memory characteristics described above have been consistently observed in bit cells fabricated in

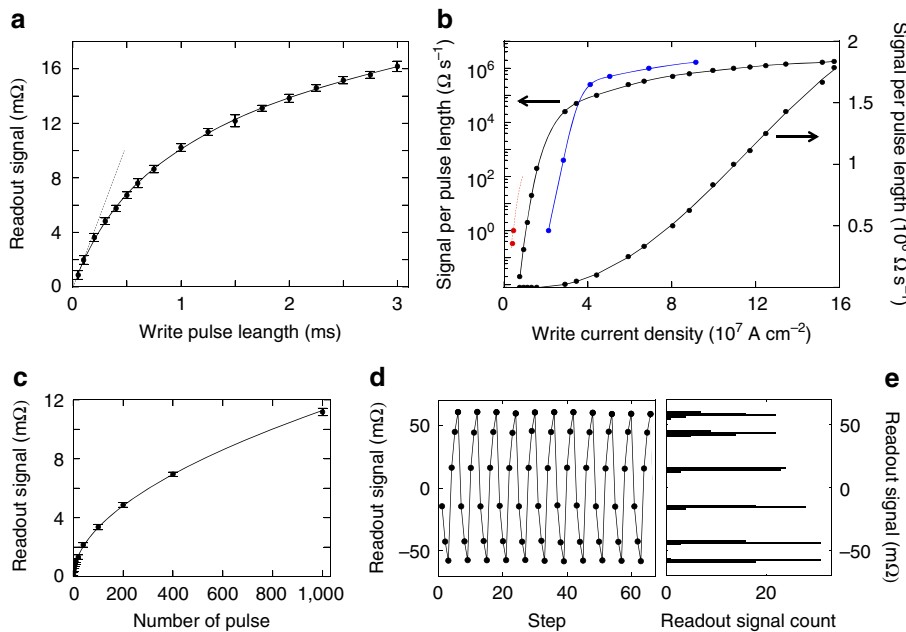

**Figure 4 | Pulse length dependence and III–V and Si compatibility. (a)** Readout signal of a 4 µm CuMnAs/GaAs device as a function of the applied write-pulse length at a fixed current density of $1.2 \times 10^7$ A cm$^{-2}$. Reading is performed with a current density of $1 \times 10^5$ A cm$^{-2}$, 5 s after the write pulse. The initial linear slope of the dependence (signal per pulse length ratio) is highlighted by the dashed line. Plotted data points are the average over fifteen measurements; error bars represent the standard deviation. **(b)** Readout signal per write-pulse length obtained from the initial linear slope (see **a**) as a function of the write current density, for 30 µm CuMnAs/GaP (red), 4 µm CuMnAs/GaAs (black), and 2 µm CuMnAs/GaP (blue) devices. **(c)** Readout signal as a function of the number of pulses in the train of pulses for the individual pulse length of 250 ps and writing current density $16 \times 10^7$ A cm$^{-2}$ in a 4 µs CuMnAs/GaAs devices. Plotted data points are the average over fifteen measurements; error bars represent the standard deviation. **(d)** Multi-level switching in the device fabricated from CuMnAs/Si. Three pulses are applied along the [100] direction followed by three pulses along the [010] direction with current density $2 \times 10^7$ A cm$^{-2}$ and pulse length 100 µm. **(e)** Histogram of the six different states, obtained from 50 repetitions of the $3+3$ pulse sequence (bin size is 1.4 m$\Omega$). All measurements were performed at room temperature.

our single-layer antiferromagnetic CuMnAs deposited at low temperature (220–300 °C) on Si or III–V substrates which opens the prospect of their utility in micro- and opto-electronics. The cells have an elementary cross-shape geometry with no intentional complexities introduced either during the layer growth or device fabrication. We have completed the excursion over the entire range of electrically generated pulses in micron-size AF bit-cells without any specific materials or microstructure optimization. It implies that these AF bit-cells should respond to still significantly reduced pulse lengths, consistent with the THz range of spin dynamics in AFs. While experiments with optically generated ultra-short pulses are beyond the scope of the present work, our complete set of electrical measurements provides the basic feasibility check and quantitative guidelines for extending the AF spin-orbit torque experiments in CuMnAs microstructures down to $\sim$10–1 ps time scales. This opens a realistic prospect of future research and applications fully exploiting the ultra-fast AF spin dynamics and the prospect of bridging the fields of spin-microelectronics with opto-spintronics in a single metallic magnetic material.

The envisaged applications of the multilevel AF bit cells are not, at least in short-term, in the non-volatile high-density computer memories. Instead, the targeted area is in specialized embedded applications for, for example, the IoT technologies. The perceived components comprise multi-level AF bit-cells where each integrates the memory-counter functionality. This concept does not impose the stringent requirements on endurance, retention, and scalability of high-density computer memories. With this in mind, we performed experiments in which the output signal was measured seconds after the last pulse in the train ($10^3$–$10^{10}$ larger times than the pulse length). This data acquisition time scale is sufficient for a range of envisaged IoT applications utilizing the embedded AF components. For some applications, a relatively fast relaxation might be even favourable since it would provide a straightforward means for resetting the AF memory-counter even without using the orthogonal writing current path and thus further simplifying the bit-cell design.

A systematic study aiming to achieve high endurance and retention properties of CuMnAs bit cells is not central to the concept of the multi-level AF memory-counter for embedded applications and is, therefore, beyond the scope of the present paper. Nevertheless, from the data accumulated to date we can comment that our ohmic devices do not suffer from fatigue, unless biased with excessively large writing currents. Regarding the retention, we observe a broad variation of relaxation times, depending on the material synthesis parameters, temperature, etc. These range from retention persisting over the entire measurement session to short relaxation times of the order of seconds. The former has been illustrated, for example, in Fig. 4 and S3 in ref. 8 and applies also to the present data on CuMnAs/Si devices. A material of the latter property was used in the measurements of up to 1,000 pulses with individual pulse length downscaled to 250 ps. The relatively short relaxation time in seconds allowed us to perform the large number of measurements with the same (for inferring error bars) or different (for assessing the counter functionality) trains of pulses without having to involve the orthogonal current path for resetting the bit-cell. In the $\sim\mu$s–ms pulse-length measurements we used a material with a longer retention time and for resetting we employed the spin-orbit field generated by the orthogonal writing current path and heating to accelerate the relaxation to the initial state. We emphasize, however, that in all these measurements the readout was performed at times exceeding the pulse length by several orders of magnitudes and relaxation, if present, had no effect on the reported results.

## Methods

**CuMnAs growth.** CuMnAs films were grown on GaAs(001), GaP(001), or Si(001) by molecular beam epitaxy at a substrate temperature of 220–300 °C. X-ray diffraction measurements showed that the films have the tetragonal $Cu_2Sb$ structure (space group P4/nmm). The film and substrate follow the epitaxial relationship CuMnAs (001)[100]||GaAs/GaP/Si(001)[110], with <1% lattice mismatch for the GaP and Si substrates. Magnetic measurements confirmed that the CuMnAs film is a compensated antiferromagnet. The Néel temperature is 480 K.

**Lithography.** Microfabrication of our CuMnAs cross-shape cells was done using electron beam lithography and reactive ion etching. We used metal masks prepared by lift-off method directly on the surface of CuMnAs films to define the pattern and protect CuMnAs layer during the reactive ion etching step. We have developed fabrication recipes enabling us to control the degree of oxidation of the CuMnAs surface below contact pads that is necessary to control the overall resistance of the devices.

**Data availability.** The datasets generated and/or analysed during this study are available from the corresponding author.

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

## Acknowledgements

We acknowledge support from the EU ERC Advanced Grant No. 268066, from the Ministry of Education of the Czech Republic Grant No. LM2015087, from the Grant Agency of the Czech Republic Grant No. 14-37427, from the University of Nottingham EPSRC Impact Acceleration Account grant No. EP/K503800/1, and from the Swiss National Science Foundation, Grant No. 200021-153404.

## Author contributions

K.O., V.S., X.M., V.N., P.W., K.W.E., B.L.G., M.B., P.G. and T.J. conceived and designed experiments. K.O., V.S. Z.K. and M.B. performed experiments. K.O., V.S., T.J. and

analysed data. K.O., X.M., V.N., Z.K., P.W., R.P.C. and J.G. contributed materials. V.S., P.G. and T.J. wrote the paper.

## Additional information

**Competing interests:** The authors declare no competing financial interests.

