## [Peer Review File · Nature Communications]

Reviewers' comments:

Reviewer #1 (Remarks to the Author):

This manuscript describes an experimental demonstration of a multi-level memory operation in an antiferromagnetic CuMnAs device driven by a relativistic spin-orbit torque. The authors expand upon their previous works [e.g., *Science*, vol. 351, p. 587 (2016)] and characterize the change in magnetization state with the applied multiple current pulses with various densities, lengths, duty cycles, and integrated times. The results are nicely presented and manuscript is well organized, and thus I basically agree that this work may open a new research avenue with information technologies at its horizon based on an entirely new type of components as the authors claims. However, I have some suggestions (listed below) which, I believe, make this manuscript further variable, and I am willing to recommend the publication if the authors satisfactorily address them.

1. I think that the authors should elaborate on the reason that the device shows multi-level operation. In page 4, the authors describe that "the antiferromagnetic moment reorientations within multiple domains of sub-100 nm dimension" cause the multi-level switching. I am wondering why some domains switches and others do not by the pulse application even though the film has almost uniform structure epitaxially grown on a single crystalline substrate. In addition, is the domain size indeed sub 100 nm? The size of domain does not look sub 100 nm in the XMLD-PEEM images presented in arXiv:1607.08478.
2. Since the author claim that the device serves as "non-volatile solid state memory (line 11 in page 2)," some results on the retention property need to be shown (or at least some comments are needed). What happens if the readout operation is repeated at a constant interval after the write pulse injection?
3. The following information should be specified unless there are any particular reasons:
 - (i) Magnitude of applied current for readout.
 - (ii) Resistance or resistivity of the CuMnAs device.
 - (iii) How to reset the cell (line 98, line 121).
4. Broken lines in Fig. 2a, Fig. 3c, Fig. 3d should be explained. Are they just eye guide, or based on analytical formula?

Reviewer #2 (Remarks to the Author):

This paper provides an interesting and thought provoking follow-up to the author's recent publication in *Science* (Ref. 3 of the manuscript) where they have shown electrical switching of antiferromagnet films and then readout using the anisotropic magnetoresistance (AMR). Their *Science* paper is a remarkable contribution to the field of spintronics and has generate remarkable interest. The argument of the current manuscript is that they further "demonstrate the complete write/store/read functionality in an antiferromagnetic CuMnAs bit cell embedded in a standard printed circuit board communicating with a computer via a USB interface" and further show a multilevel recording results. To achieve this they have used a simpler device architecture (4 leads instead of 8 leads in *Science* paper) and then scaling the size of the device down (to 2 microns on a side down from 30 microns). The device scaling allows higher current densities and the switching to occur in much faster write times which appears to be the main novelty of the work. They further focus significantly on the multilevel characteristic of the device which appears to be shown in the original *Science* article (Fig. 2E) which looks very similar to Fig. 1d and 2d of the manuscript and demonstrates a memristive property of the device and material. Much of the multilevel results use microsecond to millisecond pulse trains comparable to the original *Science* article.

The authors show that they can achieve different resistance levels and argue that their results "highlight the level of electrical control that can be achieved over these multi-domain switching processes in antiferromagnets which, unlike ferromagnets, are insensitive to and do not generate dipolar magnetic fields." I'm not quite sure how the current results are superior to other example

of ferromagnetic systems that show multilevel resistance or magnetization. In ferromagnetic materials and devices there are examples such as Y Fang, et al., *Advanced Functional Materials* 23 (15), 1919-1922 and S. Fukami, et al., *Nature Materials* 15, 535–541 (2016) Fig. 3. More generally the authors are showing what is known as a remanent curve in magnetic recording where you apply a pulse (a field in magnetic recording) and then measure the amplitude of the signal at remanence after the pulse was applied. This simply requires stable or metastable remanent domain states and device or sample size sufficiently large to sustain multiple domains. The current paper confirms the domain structure seen in Fig. 2C of their *Science* paper. This approach does not, however, scale well to smaller devices and the number of domains per device becomes smaller and there are more sample-to-sample variations. The results certainly provide valuable insight into the switching behavior and the role of Joule heating but the results do not appear, as shown, to be a particularly effective memory cell.

Based on their *Science* paper the size of the measured resistance effect is 0.2 % for the AFM AMR. Now the different levels are less than 10% of the full signal so it appears they need to resolve 0.02% changes in the resistance. This can easily be done in a single device with long measurement times. But for a memory this requires device to device resistance levels distributions to be significantly below this value. When TMR devices are running into trouble with 200% MR values it appears the AFM AMR has a long way to go to be a non-volatile memory unless the output signal is increased a 1000x.

Further the memory properties may be ultrafast in switching as stated in the article but this is not shown. The switching and time scales, as presented, are comparable with ferromagnetic devices and that the results show that heating is important make ultrafast switching seem difficult or potentially inconsistent with current measurements.

The focus of the paper that the results are done with a "standard printed circuit board communicating with a computer via a USB interface" does not seem particularly relevant. This appears to be a nice demonstration for talks but doesn't seem to advance either the science or technology.

Finally the authors state that they demonstrate write/store/read functionality. While the write and read functionality is clearly demonstrated the ability to store data requires determining an energy barrier. It seems that the results are stable but there is no discussion of thermal stability.

This group is clearly the leaders in antiferromagnetic spintronics. Because they are the leader of this new field one may make an argument that *Nature Communications* is an appropriate venue. However, it is my opinion that there is a bit too much overlap with the previous *Science* article where the authors have recast the results in terms of a memory device.

Reviewer #3 (Remarks to the Author):

This paper demonstrates the switching of an antiferromagnet using the anti-symmetric spin-orbit torque experienced by the two antiparallel antiferromagnetic sub lattices. It is experimentally demonstrated that the switching of the Neel vector can be detected using the anisotropic magnetoresistance, a.k.a the planar Hall effect. The result is interesting and the experimental results are solid. As a follow up of Ref. 3, the device scale is reduced by an order of magnitude and the switching pulse length is reduced to nanoseconds. Since the switching mechanism and the read-out scheme is exactly the same to their previous pioneering work. In order to meet the requirement in novelty of *Nat. Comm.*, I suggest the authors further analyze the device for applications, as stated in the following.

(a) How many switching cycles can a memory cell stand against the heating damage? It is mentioned in the manuscript that the decrease in size increases the current density. However, the

small size make the device less stiff against the heat generated by the switching current density up to $\sim 10^7$ A/cm². For application purposes, it is important to estimate the lifetime of the memory cell in terms of switching cycles.

(b) Another important issue is the power consumption and the data stability. Since the information is stored in the magnetic order, the AFM memory cell proposed in this paper has to suffer from the magnetic recording trilemma. The longer data lifetime corresponds to a higher energy barrier to overcome during the switching process, requiring a higher current density to switch. Thus, the balance between the power consumption and the data lifetime is crucial to estimate the application potential of the proposed scheme.

Dear Editor,

we thank the reviewers for their positive feedback on our manuscript and stimulating comments. We first address the two key points that echo in the reviewers' reports: (1) The first one regards the novelty of the present work with respect to the initial demonstration of the electrically written antiferromagnetic (AF) memory, published earlier this year in Science. (2) In the second key point the reviewers request further clarification on the application potential of the AF memory devices.

(1) *Novelty*. The work published earlier this year In Science reported the first experimental observation of electrical switching of an antiferromagnet and provided a microscopic theory analysis of the new physical phenomenon underlying this experiment in CuMnAs. The measurements were done using the full equipment range of a cryostat/magneto-transport laboratory and switching was demonstrated for writing pulses of milliseconds length. In the present manuscript, now also extended by new data and additional information in the revised text, we report (i) electrical pulsing measurements reaching the limiting timescales of ~ 100 ps and showing ~ 1000 highly reproducible states of the AF multi-level bit cell. In the present work we also report (ii) highly reproducible multi-level switching characteristics of our AF bit cells implemented in a USB proof-of-concept device operated at ambient conditions directly from a PC and by this establishing the application potential of AFs in microelectronics.

More specifically:

(i) Our elementary-shape micron-size bit cells fabricated from a single-layer AF can act as a multi-level memory-counter with highly reproducible characteristics over the entire range of electrically accessible pulse lengths spanning 8 orders of magnitude, from ~ 10 ms down to ~ 100 ps. We have not even anticipated this rapid progress when publishing a few months ago the Science paper on ~ 1 - 10 ms pulses. We are not aware of any alternative physical, materials, and microstructure basis offering a comparable microelectronic memory-counter functionality spanning the entire range from macroscopic to ~ 100 ps long pulses combined with the ability to efficiently, reproducibly and reliably record thousands of pulses.

In addition we point out that we have completed the excursion over the entire range of pulse lengths, physically accessible with electrical generation, in micron-size AF bit-cells without any specific materials or microstructure optimization. It implies that these AF bit-cells should respond to still significantly reduced pulse lengths, consistent with the intrinsic THz range of spin dynamics in AFs. Our new data also allow us to infer the pulse strength required for switching by ~ 1 - 10 ps pulses. The estimated required strength of these ultra-short pulses is feasible to achieve by opto-electrical Auston switches [see e.g. talk by J. Wunderlich at www.spice.uni-mainz.de/afm-workshop-2016-talks/] or THz fields [see e.g. talk by T. Kampfrath at www.spice.uni-mainz.de/afm-workshop-2016-talks/]. This opens a realistic prospect of future research and applications fully exploiting the ultra-fast AF spin dynamics and the prospect of bridging the fields of spin-microelectronics with

opto-spintronics in a single metallic magnetic material. These direct implications of our new measurements, stimulating future research of AF opto-spintronics, are now discussed in the revised text.

(ii) We have implemented our multi-level AF bit cells in a PCB with standard transistors and a USB-powered microcontroller for sending the write/read voltage signals. This proof-of-concept device operates at ambient conditions and also shows highly reproducible multi-level switching signals with a single readout step and no additional output data processing. The demonstration opens a defined path towards the development of standalone AF-based chips with a series of multi-level bit-cells, a multiplexer, and a standard I/O and power pin-out, directly suitable for embedded microelectronic applications. Our present proof-of-concept PCB is, therefore, not merely a nice demonstration for scientific conference talks. In our experience gained from interactions with industrial partners, it has represented the essential step, still to be performed on the side of the academic research, to substantiate and make realistic the claims on the application potential of the multi-level AF bit cells. Indeed, only after verifying the functionality on the proof-of-concept PCB, the development of the standalone AF memory-counter chips has been initiated, now in a direct collaboration with industry and with specific targeted applications in the internet of things (IoT) field. The development of the standalone chips for the embedded applications is beyond the scope of this work and further details are currently under IP embargo. However, as already pointed out above, the feasibility of this new application concept has been directly derived from the PCB proof-of-concept device introduced in the present manuscript. In the revised text we describe in more detail the circuitry of the PCB and its merit in bridging the gap between the academic research and industrial applications. We point out that having a tangible proof-of-concept demonstration is particularly important for our present case where the basic physics foundation of the concept was experimentally demonstrated only a year ago while there had been a common perception over many decades that AFs had no application potential as active parts of microelectronic devices.

(2) *Application potential.* The AF bit cells we focus on in this study and where we demonstrate up to ~ 1000 of states per bit cell are not targeting the area of non-volatile high-density main computer memories, unlike e.g. the main-stream spintronic research of ferromagnetic MRAMs. In main computer memories like MRAMs, aiming at GB densities capacity and access rates in the GB/s range, high endurance and retention stability of the bi-stable bit-cells are important particularly for minimizing the error rate. Moreover, the high endurance and retention stability has to be maintained while down-scaling the bit-cell size to the $\sim 10\text{nm}$ range of high-density IC memories. The multi-level nature of the AF bit-cells we focus on is, in general, not favorable for maximizing the retention and the bit-cell size scalability. However, as we now emphasize explicitly in the revised manuscript, the envisaged applications of the multilevel AF bit cells are not, at least in short-term, in the non-volatile high-density computer memories. Instead, the targeted area is in specialized embedded applications for e.g. the IoT technologies. The perceived components are standalone chips with a limited number of the multi-level AF bit-cells, each integrating the memory-counter

functionality. This concept does not impose the stringent requirements on endurance, retention, and scalability of high-density computer memories. (For some applications, a relatively fast relaxation might be even favorable because it would provide a straightforward means for resetting the AF memory-counter even without using the orthogonal writing current path and thus further simplifying the bit-cell design.)

A systematic study aiming to achieve high endurance and retention properties of CuMnAs bit cells is, therefore, not central to the concept of the multi-level AF memory-counter for embedded applications and is beyond the scope of the present manuscript. With this in mind, we performed experiments in which the output signal is measured seconds after the last pulse in the train, i.e., at times 10^3 - 10^{10} larger than the pulse length. This data acquisition time scale is sufficient for a range of IoT applications utilizing the embedded AF components.

Still, to address the reviewers' questions on endurance and retention within our knowledge gained from data accumulated so far, we can comment that our ohmic devices do not suffer from fatigue, unless biased with excessively large writing currents. For, e.g., the individual data points in the memory-counter measurements with up to 1000 a thousand 250ps-long pulses, the error bars were obtained from fifteen independent measurements implying that the bit cell was exposed to $\sim 25,000$ writing pulses during this study. Regarding the retention, we observe a broad variation of relaxation times, depending on the material synthesis parameters, temperature, etc. These range from no or insignificant relaxation observed during the entire measurement session to short relaxation times of the order of seconds. The former is illustrated e.g. in Figs. 4 and S3 in Wadley et al. Science 2016. A material of the latter property was used e.g. in the new measurements we now show in the revised text of up to 1000 pulses with individual pulse length downscaled to 250 ps. The relatively short relaxation time in of a few seconds allowed us to perform the large number of measurements with the same (for inferring error bars) or different (for assessing the counter functionality) trains of pulses without having to involve the orthogonal current path for resetting the bit-cell. In the s-ms measurements shown already in the original submission we employed the orthogonal current path and heating to reset the state before starting a new pulse-train measurement. We emphasize, however, that in all these measurements the readout was performed at times exceeding the pulse length by several orders of magnitudes and relaxation, if present, had no marked effect on the reported results.

All these points related to the application potential of the multi-level AF memory are now included in the revised text. To avoid any potential remaining confusion we have also removed the term "non-volatile" from the paper whose meaning may differ from context to context.

We now respond point by point to the remaining comments.

Reviewer #1 (Remarks to the Author):

Comment

1. I think that the authors should elaborate on the reason that the device shows multi-level operation. In page 4, the authors describe that “the antiferromagnetic moment reorientations within multiple domains of sub-100 nm dimension” cause the multi-level switching. I am wondering why some domains switches and others do not by the pulse application even though the film has almost uniform structure epitaxially grown on a single crystalline substrate. In addition, is the domain size indeed sub 100 nm? The size of domain does not look sub 100 nm in the XMLD-PEEM images presented in arXiv:1607.08478.

Response

Microscopic imaging of the domain reconfiguration during the switching and its correlation to the detailed structural properties of the materials are indeed beyond the scope of the present paper. All available data on this are summarized in the arXiv:1607.08478 manuscript, showing that the staggered spin-orbit fields generated by electrical current can induce modification of the antiferromagnetic domain structure. A clear correlation between the average domain orientation and the electrical readout signal is demonstrated, with both showing reproducible switching in response to orthogonally applied current pulses. However, the behavior seen in the microscopic domain images is inhomogeneous at the submicron level, highlighting the complex nature of the switching process in our multi-domain antiferromagnetic films.

Following this reviewer’s comment we replaced in the revised text “sub 100 nm” with “sub micron” domain dimensions which indeed better correlates with the microscopic XMLD-PEEM images. We also note that our initial studies indicate a dependence of the domain size on the thickness of the CuMnAs films and substrate. In 10nm and thinner films we observe large domains exceeding 10micron dimensions. A systematic study of films with different domain sizes will be published elsewhere.

Comment

2. Since the author claim that the device serves as “non-volatile solid state memory (line 11 in page 2),” some results on the retention property need to be shown (or at least some comments are needed). What happens if the readout operation is repeated at a constant interval after the write pulse injection?

Response

This comment is addressed above in the 3rd paragraph of point (2) *Application potential* of the common part of this response and the manuscript has been revised accordingly.

Comment

3. The following information should be specified unless there are any particular reasons:
(i) Magnitude of applied current for readout.

(ii) Resistance or resistivity of the CuMnAs device.

Response

The readout current was 500A and the CuMnAs conductivity is $8 \times 10^{3-1} \text{cm}^{-1}$. This is now included in the revised text.

Comment

(iii) How to reset the cell (line 98, line 121).

Response

This comment is addressed above in the 3rd paragraph of point (2) *Application potential* of the common part of this response and the manuscript has been revised accordingly.

Comment

4. Broken lines in Fig. 2a, Fig. 3c, Fig. 3d should be explained. Are they just eye guide, or based on analytical formula?

Response

These are guide to the eye curves as we now also mention in the revised figure captions.

Reviewer #2 (Remarks to the Author):

Comment

The authors show that they can achieve different resistance levels and argue that their results “highlight the level of electrical control that can be achieved over these multi-domain switching processes in antiferromagnets which, unlike ferromagnets, are insensitive to and do not generate dipolar magnetic fields.” I’m not quite sure how the current results are superior to other example of ferromagnetic systems that show multilevel resistance or magnetization. In ferromagnetic materials and devices there are examples such as Y Fang, et al., *Advanced Functional Materials* 23 (15), 1919-1922 and S. Fukami, et al., *Nature Materials* 15, 535–541 (2016) Fig. 3. More generally the authors are showing what is known as a remanent curve in magnetic recording where you apply a pulse (a field in magnetic recording) and then measure the amplitude of the signal at remanence after the pulse was applied. This simply requires stable or metastable remanent domain states and device or sample size sufficiently large to sustain multiple domains. The current paper confirms the domain structure seen in Fig. 2C of their *Science* paper. This approach does not, however, scale well to smaller devices and the number of domains per device becomes smaller and there are more sample-to-sample variations. The results certainly provide

valuable insight into the switching behavior and the role of Joule heating but the results do not appear, as shown, to be a particularly effective memory cell.

Response

This comment is addressed above in the section (i) of point (1) *Novelty* and in point (2) *Application potential* of the common part of this response and the manuscript has been revised accordingly.

Comment

Bases on their Science paper the size of the measured resistance effect is 0.2 % for the AFM AMR. Now the different levels are less than 10% of the full signal so it appears they need to resolve 0.02% changes in the resistance. This can easily be done in a single device with long measurement times. But for a memory this requires device to device resistance levels distributions to be significantly below this value. When TMR devices are running into trouble with 200% MR values it appears the AFM AMR has a long way to go to be a non-volatile memory unless the output signal is increased a 1000x.

Response

The direct feasibility of implementing our bit-cells in embedded memory-counter applications is addressed above in the section (ii) of point (1) *Novelty* of the common part of this response. The envisaged applications are then discussed in point (2) *Application potential* of the common part of this response. The manuscript has been revised accordingly.

Comment

Further the memory properties may be ultrafast in switching as stated in the article but this is not shown. The switching and time scales, as presented, are comparable with ferromagnetic devices and that the results show that heating is important make ultrafast switching seem difficult or potentially inconsistent with current measurements.

Response

This is addressed by our new measurements discussed in the section (i) of point (1) *Novelty* of the common part of this response and the manuscript has been revised accordingly.

Comment

The focus of the paper that the results are done with a “standard printed circuit board communicating with a computer via a USB interface” does not seem particular relevant. This appears to be a nice demonstration for talks but doesn’t seem to advance either

the science or technology.

Response

We agree with the reviewer that such a demonstrator does not advance the science, but it does show that AF materials can be used as a data storage medium using very simple peripheral electronics. We believe that this is an essential step to stimulate further interest in AF spintronics research and applications. In more detail, this comment is addressed above in the section (ii) of point (1) *Novelty* of the common part of this response and the manuscript has been revised accordingly.

Comment

Finally the authors state that they demonstrate write/store/read functionality. While the write and read functionality is clearly demonstrated the ability to store data requires determining an energy barrier. It seems that the results are stable but there is no discussion of thermal stability.

Response

This comment is addressed above in the first three paragraphs of point (2) *Application potential* of the common part of this response and the manuscript has been revised accordingly.

Comment

This group is clearly the leaders in antiferromagnetic spintronics. Because they are the leader of this new field one may make an argument that Nature Communication is appropriate venue. However, it is my opinion that there is a bit too much overlap with the previous Science article where the authors have recast the results in term of a memory device.

Response

This comment is addressed above in sections (i) and (ii) of point (1) *Novelty* of the common part of this response and the manuscript has been revised accordingly.

Reviewer #3 (Remarks to the Author):

Comment

(a) How many switching cycles can a memory cell stand against the heating damage? It is mentioned in the manuscript that the decrease in size increases the current density. However, the small size make the device less stiff against the heat generated by the switching current density up to $\sim 10^7$ A/cm². For application purposes, it is important to estimate the lifetime of the memory cell in terms of switching cycles.

Response

This comment is addressed above in the 3rd paragraph of point (2) *Application potential* of the common part of this response and the manuscript has been revised accordingly.

Comment

(b) Another important issue is the power consumption and the data stability. Since the information is stored in the magnetic order, the AFM memory cell proposed in this paper has to suffer from the magnetic recording trilemma. The longer data lifetime corresponds to a higher energy barrier to overcome during the switching process, requiring a higher current density to switch. Thus, the balance between the power consumption and the data lifetime is crucial to estimate the application potential of the proposed scheme.

Response

This comment is addressed above in point (2) *Application potential* of the common part of this response and the manuscript has been revised accordingly.

Sincerely,

Kamil Olejnik and Tomas Jungwirth in the name of all co-authors

Reviewers' comments:

Reviewer #1 (Remarks to the Author):

The revised manuscript is written in such a way that experimental results on the dependence of change in magnetization state on pulse length, and so on are presented first and then application potential is described in detail. The revision has been mainly made on the second part. Because I thought that the discussion on underlying physics behind the "Antiferromagnetic multi-level memory cell" described in the first part was insufficient, I requested the authors to elaborate on this point at the first round. However, their response is that, shortly speaking, it is beyond the scope of this manuscript. Instead, the authors revised their manuscript so that the feasibility of the antiferromagnetic multi-level memory cell applied to IoT fields is emphasized. If we regard this device as a viable candidate for such application, I think the device has some drawbacks at least in the following points: (1) usage of MBE and expensive single crystalline substrate with elevated temperature for film growth, (2) narrow temperature range in which the device works without any change in function, (3) output resistance (or voltage) change, which is currently in the order of mOhm (mV), (4) device structure that requires four terminals, and so on. In my opinion, discussion on these points is also necessary to satisfy the criteria of Nature Communications. I also note that the authors should compare their proposed device with other solid-state devices and circuits including conventional CMOS based technologies as well as with ferromagnetic devices, and indicate some advantages. CMOS based integrated circuits should be able to easily and more reliably implement the function that the antiferromagnetic multi-level memory cell possesses, and other oxide, ferroelectric, and phase-change materials based devices may also be competitors. I think that the experimentally observed phenomena themselves are very interesting and the readers of Nature Communications should enjoy this manuscript if the physics to support the results, such as the reason for multilevel state including microscopic pictures, dynamics of activation/relaxation in connection with the Joule heating, and future issues, are sufficiently described. Perhaps, combining this manuscript with their arXiv: 1607.08478 might be a better way. On the other hand, if the author wish to publish their manuscript with showing just what they saw and discussing the feasibility for application in detail, they also have to address the above issues so that the broad audience can accept. Overall, I highly acknowledge that the results shown in their previous Science paper are ground-breaking in the history of the research on antiferromagnet, and the results shown in the present manuscript are also a significant step at least in the field of magnetics/spintronics. However, as far as the experimental results is just shown without satisfactorily discussion and the application potential is enthusiastically emphasized, which looks to me insufficient, I regret to say that I cannot still recommend the manuscript for publication in Nature Communications.

The following are additional minor comments/questions.

- (i) Since no supporting data is shown for "opt-spintronics", I think the authors should not use the term in this manuscript (at least in the abstract).
- (ii) "temperature during switching stays at least 100 K (line 14 in page 6)" should be "temperature increase during ..."

Reviewer #2 (Remarks to the Author):

I stand by many of the my comments of the original review, that this paper provides an interesting and thought provoking follow-up to the author's recent publication in Science but doesn't seem to be well suited to Nature Communications.

Regarding some specific aspects of the revised paper and response. Comment (ii) of the response is about the PCB and USB-powered microcontroller. I went to some of the talks at the afm-workshop that the authors highlighted in their response. One of the authors specifically discusses the PCB implementation. Now maybe they were trying to be entertaining but they specifically say in the talk that the implementation on the PC was good for highlighting the work for press releases

and to sell it to investors. At one point the author states in their talk "Why we need a USB gadget? When you don't have a USB gadget, and you publish a paper the news are featuring a chip carrier or a cryostat". The author goes on to say that press coverage is better with the gadget and it would be even better with a blinking light. The author further goes on to say that there is a direct relation between the successes of a start-up with a tangible devices. So I stand by statement that this implementation does little to move either the science or fundamental technology that would be appropriate in Nature Communications but is a bit of showmanship.

The authors responded, in part, to my comments on the size of the measured resistance effect of 0.2 % for the AFM AMR. They have revised the manuscript to now focus on a specific application as a memory-counter with a large signal, scalability nor stability are important. This seems to be a bit more niche application. However, even in this application a 0.2% is very small where you want to detect 1000 different levels. I would expect that even if there were temperature changes of 10 - 20 K you would get such resistance changes. Similarly are the levels achieved from device to device the same?

I still generally agree with the concluding paragraph of my first report: "This group is clearly the leaders in antiferromagnetic spintronics. Because they are the leader of this new field one may make an argument that Nature Communication is appropriate venue. However, it is my opinion that there is a bit too much overlap with the previous Science article where the authors have recast the results in term of a memory device." I still believe this although I would certainly understand an Editorial decision to accept this paper.

Reviewer #3 (Remarks to the Author):

The revised manuscript and the authors' response clearly state the stability and the life time of the proposed devices.

Although relaxation does exist in the proposed data bits, it does not affect the validity of the observations.

These facts clearly demonstrate the application potential of the proposed device, and thus represent a significant technological leap in AFM spintronics. I recommend publication of the paper in Nat. Comm.

Dear Editor,

we thank the reviewers for their positive feedback. Below find please our response to the remaining comments of Reviewer #1 which we have used to improve the discussion of the microscopic origin of our results and of their potential future applications. Changes in the revised manuscript are highlighted in red for clarity.

Before addressing the specific comments of Reviewer #1 we first recall the new experimental results added in the previous revision that are relevant for our responses below and highlight new experiments that we included in this round.

In the previous revision we included an extensive set of new measurements of more than 25,000 switchings and with individual pulse length downscaled to the ~ 100 ps time scale. Combined with the data from the original version, we demonstrated that our elementary-shape micron-size bit cells can act as a multi-level memory-counter over the entire range of pulse lengths accessible by electrical generation, spanning 8 orders of magnitude from ~ 10 ms down to ~ 100 ps. The completion of this excursion over the full range of electrically generated pulses did not require any specific materials or microstructure optimization. This is very promising when considering potential future applications in microelectronics and it also opens a realistic prospect of extending the functionality of our CuMnAs bit cells towards the ps-timescale of the internal spin dynamics in the AF.

In this revision we include yet another experimental breakthrough: We have successfully grown our CuMnAs films at low temperature (220°C) on a Si substrate. Devices fabricated from these films show the highly reproducible deterministic multi-level switching as previously observed in CuMnAs films on III-V substrates. The figure below shows a comparison of the earlier data on CuMnAs/GaP with the corresponding newly replaced panel in Fig. 4 featuring the data on CuMnAs/Si. We regard the successful demonstration of CuMnAs/Si devices a major step towards a broad applicability of the antiferromagnetic memory-logic concept in microelectronics.

We now proceed to the point-by-point response to the specific comments:

Comment

Because I thought that the discussion on underlying physics behind the “Antiferromagnetic multi-level memory cell” described in the first part was insufficient, I requested the authors to elaborate on this point at the first round. However, their response is that, shortly speaking, it is beyond the scope of this manuscript... Perhaps, combining this manuscript with their arXiv: 1607.08478 might be a better way.

Response

We agree with the Reviewer on the importance of understanding the microscopic picture of the multi-level switching characteristics of the CuMnAs devices. Following the Reviewer's recommendation, we have extended the discussion of this point borrowing from the conclusions of Ref. 13 (originally arXiv:1607.08478 now in press in Phys. Rev. Lett.). In the revised text we also note the distinction between the type of the study reported in Ref. 13 and our experiments discussed in the present manuscript. We have added the following text in the revised manuscript:

A complementary study performed at the Diamond Light Source directly associated the electrical switching signal in a CuMnAs cross structure with 10 m wide arms with the AF moment reorientations within multiple domains of sub-micron dimensions.¹³ In the experiment, several pairs of orthogonal, 50 ms writing pulses were applied and the corresponding domain reconfigurations were detected by means of the photoemission electron microscopy (PEEM) with contrast enabled by x-ray magnetic linear dichroism (XMLD). The observed spatially-averaged XMLD-PEEM signal correlated well with the measured AMR which also represented a magnetoresistance signal averaged over many domains. On a sub-micron scale, however, the XMLD-PEEM images showed a non-uniformity with domains responding significantly stronger or weaker to the writing pulses than the spatial average. Consistently, when several successive writing pulses were applied along the same direction, the increasing AMR signal of the multi-level bit cell again correlated well with the increased number of switched domains as observed in the XMLD-PEEM.¹³

While in the large-facility XMLD-PEEM experiment only a very limited number of switchings could have been explored, Figs.~2c,d highlight on hundreds of pulses the level of electrical control that can be achieved over the multi-domain switching processes in AFs which, unlike ferromagnets,¹⁹ are insensitive to and do not generate dipolar magnetic fields. We now proceed to exploring in detail the dependencies of the readout signals on the parameters of writing pulses. The study presented below involves tens of thousands of switchings with individual pulse lengths spanning eight orders of magnitude from ~10 ms down to ~100 ps. We performed the experiments using laboratory electrical pulse generators or high frequency set-ups equipped with rf cables and the AF devices mounted on specially designed co-planar waveguide with rf access.

Comment

If we regard this device as a viable candidate for such application,

Response

It was not our intention for the USB demonstrator device itself to be regarded as a viable candidate for an IoT application. The aim of our demonstrator device is to provide a tangible, proof-of-concept support for our discussion of potential future applications of multi-level AF memories. We consider this intermediate step between science and technology highly relevant, in particular for AFs that have been perceived for nearly a century as useless for practical applications. This intermediate step is our attempt to build the bridge over the valley of death between academia and industry, not to substitute the development and marketing activities on the industry side of the valley.

To clarify this point in the revised text, we have added the word **future** when referring to IoT applications. In the Abstract we now state that: **Our results open a path towards specialized embedded memory applications of antiferromagnets and towards ultra-fast memory components...** In the Discussion we state that **...this work opens a defined path towards the development...**

Comment

... I think the device has some drawbacks at least in the following points: (1) usage of MBE and expensive single crystalline substrate with elevated temperature for film growth

Response

To address this point we have included the following information/clarification in the revised text when introducing our materials:

In the present paper we focus on the multi-level switching characteristics of the memory bit-cells patterned into an elementary cross-shape geometry from a single metallic layer of the CuMnAs AF deposited on a III-V or Si substrate....

Figures 1 and 2 provide an overview of the basic characteristics of our AF CuMnAs memory cells. For the purpose of the present study the cell has a cross shape, 2 μm in size (Fig.~1b), patterned by electron beam lithography and reactive ion etching from a 60-nm thick, single-crystal CuMnAs film (Fig.~1a). The material was grown by molecular beam epitaxy (MBE) on a GaP substrate.¹⁵ We recall that, besides basic research, MBE is widely used in the manufacture of microelectronic devices, in particular for mobile technologies. We also note that GaP is lattice matched to Si and that, as shown below, high quality CuMnAs films can be deposited on both GaP and Si at temperatures between 220 and 300°C, i.e., well below the CMOS circuit tolerance limit which is typically above 400°C....

Finally we illustrate in Fig.~4d that bit-cells fabricated from CuMnAs films

deposited on Si at 220°C also show the highly-reproducible multi-level switching characteristics as the devices fabricated from CuMnAs on GaP or GaAs substrates (cf. with Fig. 2 and Ref. 8). The plot shows an example of a symmetric pulsing experiment of repeating three writing pulses with current lines along the [100] direction followed by three pulses with current lines along the [010] direction. The corresponding histogram taken from 300 pulses highlights the deterministic switching of these multi-level CuMnAs/Si bit-cells.

We add to this here that, while MBE is not a general show-stopper in applications, our demonstrator device is not on the level of technological development that would allow us to make a credible costing analysis for a specific commercial product. Also, other more cost-efficient growth techniques might be applicable to CuMnAs or alternative AF materials and we expect intense materials research in this direction, in analogy to the development of ferromagnetic spintronic devices.

Comment

(2) narrow temperature range in which the device works without any change in function, (3) output resistance (or voltage) change, which is currently in the order of mOhm (mV)

Response

To address these points we have included the following clarification in the revised manuscript:

We note that ohmic anisotropic magnetoresistance of comparable magnitude to our CuMnAs films¹³ was utilized in the first generation of MRAM integrated circuits using thin-film uniaxial ferromagnets and bridge formation in the read circuitry, comprising reference and storing cells, for eliminating thermal and noise effects.^{11,17,18}

Comment

(4) device structure that requires four terminals

Response

Ref. 11 in the above clarification describes also the details of the concept of the AMR readout in a uniaxial AF which, in principle, requires only two terminals. The concept again borrows from the previous realizations of ferromagnetic AMR MRAMs in which the bit cells were not fabricated from biaxial but uniaxial ferromagnets, as pointed out explicitly in the above clarification.

Comment

I also note that the authors should compare their proposed device with other solid-state devices and circuits including conventional CMOS based technologies as well as with ferromagnetic devices, and indicate some advantages. CMOS based integrated circuits should be able to easily and more reliably implement the function that the antiferromagnetic multi-level memory cell possesses, and other oxide, ferroelectric, and phase-change materials based devices may also be competitors.

Response

To address this point we have added the following in the revised text when first introducing the IoT:

However, in combination with the simplicity of the bit-cell geometry and unique features of AFs stemming from their zero net moment, the multi-level nature may provide additional functionalities, such as a pulse counter, with a utility in future specialized embedded memory-logic components in the "More than Moore"¹⁴ internet of things (IoT) applications.

We have added here the new Ref. 14 to a recent review article describing that the era of the International Technology Roadmap for Semiconductors is officially at an end, and that an entirely new approach is now being formulated under the International Roadmap for Devices and Systems. Briefly summarizing from Ref. 14, the simple landscape of computer processors and memories, where benchmarking of one technology against the other has been relatively straightforward, ceased to exist with the end of the Moore's Law and with the invasion of mobile, IoT, and cloud technologies (Internet of Everything). As described in Ref. 14, it is now broadly acknowledged that the IT landscape is becoming highly fragmented and exceedingly complex, making a generic benchmarking exercise close to impossible. As we have emphasized above, our research results, including the USB proof-of-concept device, are not at the stage of targeting a specific IoT application. Instead, we demonstrate that a simple cross-geometry bit cell alone can in principle count and record thousands of pulses of lengths spanning 8 orders of magnitude down to the ~100ps range, and argue that from the physical limitation perspective downscaling the pulse lengths by another two orders of magnitude to the picosecond range should be readily feasible. To the best of our knowledge, there is no other charge or spin based concept offering anything similar to this and we have demonstrated it at ambient conditions on a material deposited at low temperature on Si or III-V substrates.

Comment

(i) Since no supporting data is shown for "opt-spintronics", I think the authors

should not use the term in this manuscript (at least in the abstract).

Response

We have removed the term opto-spintronics from the abstract.

Comment

(ii) “temperature during switching stays at least 100 K (line 14 in page 6)” should be “temperature increase during ...”

Response

The original sentence was correct. The absolute temperature stays at least 100 K below the Néel temperature.

Sincerely,

Kamil Olejnik and Tomas Jungwirth in the name of all co-authors

REVIEWERS' COMMENTS:

Reviewer #1 (Remarks to the Author):

In the revised manuscript, the authors reasonably address the points I commented at the second review round. In particular, they added results for the device deposited on Si substrate and also added some discussion on the reason for multi-level operation based on the XMCD-PEEM observation. I think the paper now satisfies the criteria of Nature Communications and thus support the publication of this manuscript. I have a minor suggestion. I thought that the switching experiment is performed at room temperature but this seems to be incorrect. I think the authors should clearly state the set temperature for each measurement.

Referee:

In the revised manuscript, the authors reasonably address the points I commented at the second review round. In particular, they added results for the device deposited on Si substrate and also added some discussion on the reason for multi-level operation based on the XMCD-PEEM observation. I think the paper now satisfies the criteria of Nature Communications and thus support the publication of this manuscript. I have a minor suggestion. I thought that the switching experiment is performed at room temperature but this seems to be incorrect. I think the authors should clearly state the set temperature for each measurement.

Response:

We have added a note on temperature into captions of all figures.